# Novel Experimental Mouse Model to Study Malaria-Associated Acute Kidney Injury

**DOI:** 10.3390/pathogens12040545

**Published:** 2023-04-01

**Authors:** Johanna Bensalel, Alexandra Roberts, Kiara Hernandez, Angelica Pina, Winifred Prempeh, Blessing V. Babalola, Pablo Cannata, Alberto Lazaro, Julio Gallego-Delgado

**Affiliations:** 1Department of Biological Sciences, Bronx, Lehman College, The City University of New York, New York, NY 10468, USA; 2Ph.D. Program in Biology, The Graduate Center, The City University of New York, New York, NY 10016, USA; 3Department of Pathology, IIS-Fundación Jiménez Díaz, School of Medicine, Universidad Autónoma de Madrid, 28040 Madrid, Spain; 4Renal Physiopathology Laboratory, Department of Nephrology, Instituto de Investigación Sanitaria Gregorio Marañón, Hospital General Universitario Gregorio Marañón, 28007 Madrid, Spain; 5Department of Physiology, School of Medicine, Universidad Complutense de Madrid, 28040 Madrid, Spain

**Keywords:** malaria, acute kidney injury, severe malaria, MAKI, severe malaria mouse model, renal impairment, AKI

## Abstract

The impact of malaria-associated acute kidney injury (MAKI), one of the strongest predictors of death in children with severe malaria (SM), has been largely underestimated and research in this area has been neglected. Consequently, a standard experimental mouse model to research this pathology is still lacking. The purpose of this study was to develop an in vivo model that resembles the pathology in MAKI patients. In this study, unilateral nephrectomies were performed on wild-type mice prior to infection with *Plasmodium berghei* NK65. The removal of one kidney has shown to be an effective approach to replicating the most common findings in humans with MAKI. Infection of nephrectomized mice, compared to their non-nephrectomized counterparts, resulted in the development of kidney injury, evident by histopathological analysis and elevated levels of acute kidney injury (AKI) biomarkers, including urinary neutrophil gelatinase-associated lipocalin, serum Cystatin C, and blood urea nitrogen. Establishment of this in vivo model of MAKI is critical to the scientific community, as it can be used to elucidate the molecular pathways implicated in MAKI, delineate the development of the disease, identify biomarkers for early diagnosis and prognosis, and test potential adjunctive therapies.

## 1. Introduction

In 2021, there were an estimated 247 million cases of malaria resulting in 619,000 deaths, with 80% of those deaths being of children under 5 years of age [1]. Approximately 40–60% of severe malaria (SM) cases present with acute kidney injury (AKI) [2,3]. Malaria-associated AKI (MAKI) increases patients’ risk of mortality, renal failure, chronic kidney disease, and other comorbidities [2,4,5,6].

Until recently, it was thought that MAKI only affected adult patients with SM, and the impact on children was largely underestimated. Recent reports, however, indicate that it does indeed occur in children with SM and is actually one of the strongest indicators of death [7,8]. This SM complication is also associated with long-term neurocognitive deficits, behavioral problems, and reduced lifespan in children [2,9]. Despite the severity of MAKI, very little is known regarding its molecular pathogenesis or clinical pathology, ultimately resulting in a lack of adjunctive therapies to treat this condition or reliable biomarkers for diagnosis. Treatments are limited to renal replacement therapy or fluid replacement when feasible, and the biomarkers normally used to define AKI (increase in serum creatinine and/or decreased urine output) may not be applicable in the case of this particular pathology. An increase in serum creatinine may not occur until advanced stages of kidney injury, and a baseline creatinine level can be difficult to establish in settings where MAKI most occurs [10].

Histopathological evaluation of renal tissue has demonstrated that acute tubular necrosis (ATN) and, to a lesser degree, glomerulonephritis and interstitial nephritis, occur in patients with MAKI [11,12,13]. There are some proposed causes for this phenomenon, including hemodynamic perturbations, oxidative stress, immune-mediated injury, and metabolic dysfunction [14,15,16]. While there is some evidence for each of these proposed causes, there is still relatively little research on the exact causes and mechanisms underlying the damage observed in MAKI as its impact was previously underestimated. The lack of basic research on this specific complication of malaria is in part due to the lack of suitable models of MAKI. The mouse models used to study this SM complication thus far have not succeeded in reproducing the pathology observed in human patients [17,18] or present other limitations, such as the development of additional SM complications [19], not having included both male and female mice [20,21], or models that were not physiological as genetically modified mice were used [22,23].

Here, we introduce an experimental mouse model of MAKI that exhibits signs of early stages of MAKI. Both male and female mice were included in this study and mice did not develop signs of other SM complications. For this model, one kidney was surgically removed from wild-type mice prior to infection with *Plasmodium berghei* NK65-NY strain (*Pb*NK65-NY). The rationale behind this study was that removal of one kidney should increase the strain from infection on the remaining kidney without affecting overall renal function. MAKI was assessed by histopathological analysis, quantification of biomarkers of AKI, and detection of proteins associated with AKI in urine and renal tissue samples. This model could address the gap in the field of MAKI research and be used to explore the development of this specific complication of malaria and its underlying mechanisms.

## 2. Results

The experimental mouse model of MAKI was performed in C57BL/6 J mice and included four groups: non-infected control mice (2KNI), mice that underwent nephrectomies but were not infected (1KNI), and their infected counterparts, non-nephrectomized (2KI) and nephrectomized (1KI) infected mice. Mice in the infected groups (2KI and 1KI) were monitored for parasitemia levels by analyzing tail vein blood smears every other day starting from day 3 post-infection until termination. Parasitemia levels rose comparably in both groups (Figure 1) indicating that the surgery did not affect the parasite’s growth pattern.

To monitor for signs of oligoanuria, urine was collected prior to infection and every 2–3 days post-infection to measure urine output. Urine samples were also collected to measure levels of kidney injury biomarkers. There were no significant differences in the volume of urine output between any of the groups on any day post-infection (Figure 2A). Urinary neutrophil gelatinase-associated lipocalin (uNGAL), a biomarker of AKI associated with MAKI [10], was measured, and levels were significantly increased in infected nephrectomized mice (1KI) compared to control non-infected mice at days 5, 8, 11, and 14 post-infection. By day 11 post-infection, 1KI uNGAL levels were significantly higher than those of mice in all other groups and remained so until termination (Figure 2B). None of the animals included in the experimental model developed signs of proteinuria as shown by the urinary albumin levels measured using the Exton method (Figure 2C).

Renal tissue and serum were collected upon termination of the model for histological analysis and to measure levels of AKI-specific biomarkers in blood. Blood urea nitrogen (BUN) levels were significantly higher only in nephrectomized infected animals (1KI group) (*n* = 10, mean = 45.07 ± 3.398) compared to the control mice (2KNI group) (*n* = 15, mean = 29.21 ± 1.416; *p* = 0.0118). Mice in the 1KNI (*n* = 11, mean = 39.06 ± 2.687) or 2KI (*n* = 10, mean = 34.93 ± 2.293) groups did not exhibit significantly higher BUN levels (Figure 3A). Serum Cystatin C (sCysC), a serum biomarker whose increase has been demonstrated in MAKI patients [9], was also significantly higher in 1KI mice (*n* = 10, mean = 724.2 ± 81.62) compared to mice in all other groups (2KNI: *n* = 15, mean = 370.4 ± 23.15, *p* < 0.0001; 1KNI: *n* = 11, mean = 411.6 ± 48.77, *p* = 0.0012; 2KI: *n* = 10, mean = 459.3 ± 61.63, *p* = 0.0090) (Figure 3B). Serum creatinine levels were also measured. The mean serum creatinine level for the 1KI group (*n* = 10, mean = 0.3333 ± 0.0558) was 1.4-fold higher than that of the 2KNI group (*n* = 14, mean = 0.2391 ± 0.052), 3-fold higher than that of the 1KNI group (*n* = 9, mean = 0.1071 ± 0.050), and 1.8-fold higher than that of the 2KI group (*n* = 10, mean = 0.1826 ± 0.0553) but this increase did not reach statistical significance (*p* = 0.0588) (Figure 3C). Additionally, there were no significant differences between any of the groups for creatinine clearance (Figure 3D).

Histological analysis of 4 µm paraffin-embedded whole renal tissue sections stained with hematoxylin-eosin were evaluated in a blind session by an experienced nephropathologist (Figure 4A). Glomerular and tubular injury were individually scored using a semi-quantitative method using an ascending scale ranging from 0–12. Infected-nephrectomized mice (1KI) received the highest scores on average for both glomerular and tubular damage. The mean total score for tubular injury of 1KI mice (*n* = 10, mean = 2.4) was significantly higher than that of the 2KNI (*n* = 15, mean = 0.0; *p* < 0.0001) and 1KNI (*n* = 11, mean = 0.5; *p* = 0.0027) mice. The mean total score for 2KI mice was half (*n* = 9, mean = 1.2) of that observed in the infected-nephrectomized mice but significantly higher than that of the 2KNI group (*p* = 0.0169), as well (Figure 4B). Similarly, in the evaluation for glomerular injury, mice in the 1KI group (*n* = 10, mean = 2.60) had significantly higher scores than those in the 2KNI (*n* = 15, mean = 0.0; *p* < 0.0001) and 1KNI (*n* = 11, mean = 0.0; *p* = 0.0013) groups (Figure 4C). The mean total score for 2KI mice (*n* = 9, mean = 1.4) was also significantly higher than that of the 2KNI mice (*p* = 0.0028). These instances of tubular and glomerular damage occurred in the cortex region, while no such remarkable instances of damage were found in the outer or inner medulla regions in any group (Figure 5). Interestingly, none of the mice included in the study exhibited signs of chronic renal damage defined by the presence of fibrous crescents, tubular atrophy, or interstitial fibrosis.

Expression of the apoptosis marker, cleaved caspase-3, was also evaluated in renal tissue via immunoblotting. Relative expression was significantly higher in mice in the infected groups versus those in the control non-infected groups (1KI *p* = 0.0014; 2KI *p* = 0.0060) (Figure 6).

## 3. Discussion

Only recently has the prevalence of MAKI in SM patients, and its severe impact on the outcome of the disease, been realized. As such, basic research on MAKI is scarce. While there are accepted standard mouse models to study other manifestations of SM [24] (e.g., cerebral malaria, severe anemia, etc.), this is not the case for MAKI. In this study, we introduce a novel experimental mouse model of MAKI using wild-type mice that can be used to investigate the renal pathology of this SM complication.

Previous attempts at replicating MAKI in vivo using mice have been made but in many of these studies animals did not display signs characteristic of MAKI, such as tubular damage or glomerulonephritis [17,18]. Some studies were successful in showing renal damage characteristic of MAKI but animals concomitantly developed other SM complications, such as acute respiratory distress syndrome or cerebral malaria, creating a situation with too many confounding variables and making it difficult to unravel the mechanisms specifically underlying the development of MAKI [19,20,21]. In other cases, studies involved knocking out genes essential for heme detoxification, demonstrating the importance of those genes in AKI pathology, but resulting in a non-physiological model of renal damage induced by malaria [22,23].

In this study, unilateral nephrectomies were performed on C57BL/6 J mice in order to increase the strain from infection on the remaining kidney. Although C57BL/6 J mice are relatively resistant to kidney damage [25], they have an advantage over other mouse strains in that, similarly to humans, they carry only one renin gene and do not develop glomerulosclerosis upon unilateral nephrectomy as other strains with two active copies of the gene do [26,27]. The *Pb*-NK65/NY strain was selected for infection so that mice would not develop other SM complications, such as cerebral malaria from infection with *P. berghei* ANKA, or malaria-acute respiratory distress syndrome from infection with *P. berghei* NK65/Edinburgh [28]. This would ensure that mice do not die from other SM complications before AKI might occur and it would also limit confounding variables in order to generate a clearer picture of the renal pathology.

There are several biomarkers that have been used as surrogate markers of MAKI [6,10,29]. In this study, we monitored urine output at three-day intervals, and measured albumin excretion rates as well as urinary NGAL levels in order to assess the progression of kidney disease. We did not see any signs of oliguria/anuria in any of the groups included in the experimental model (Figure 2A) which is in contrast to what is found in adult MAKI patients, with 45–75% of the cases presenting oligo-anuria [30], but is in accordance with what has been described in pediatric MAKI patients, of which up to 80% present with non-oliguric AKI [14]. The fact that oliguria is one of the risk factors associated with poor outcome in MAKI adult patients may indicate that this murine model mimics early stages of human MAKI. Moreover, none of the animals exhibited signs of proteinuria (Figure 2C) which is a criterion also associated with more advanced stages of acute kidney disease, or even chronic kidney disease [31]. In accordance with these findings, the urinary levels of NGAL, which is considered an early biomarker of AKI [10], were significantly elevated in animals in the infected groups, 2KI and 1KI (Figure 2C). Interestingly, at day 5, both infected groups showed significantly higher levels of uNGAL vs. non-infected groups; however, by day 11, those animals in the infected and non-nephrectomized group (2KI) recovered normal levels of uNGAL, and only the mice included in the infected and nephrectomized group (1KI) maintained uNGAL levels that were elevated four-fold compared to the rest of the groups. This could be an indication that the kidney repair capacity of mice overcomes the damage caused by the infection, and levels of uNGAL are only elevated during a very limited time. However, when mice undergo lateral nephrectomy, the remnant kidney repair capacity does not seem sufficient enough to compensate for malaria insult, and the damage, along with elevated uNGAL levels, are sustained over time. It is noteworthy to mention that the parasite load did not play a significant role in this phenomenon since both infected groups coursed with the same levels of parasitemia (Figure 1).

At endpoint, three standard serum biomarkers of renal injury were also analyzed, namely serum creatinine, BUN, and serum Cystatin C. The AKI-specific serum biomarkers, BUN and sCysC, were significantly increased in the sera of 1KI mice compared to mice in all other groups, suggesting that infected mice with two kidneys were able to manage the insult and limit the renal damage, as suggested by the uNGAL levels as well (Figure 3A,B). Serum creatinine levels were increased in 1KI mice, compared to mice in other groups, though this increase did not reach statistical significance (Figure 3C). Although increased serum creatinine levels are still used to define clinical AKI, there is growing evidence that this may be an unreliable marker in the case of malaria infection, as it may occur in late stages of kidney injury or not at all [10].

Histological analysis confirmed that 1KI mice had greater instances of tubulointerstitial damage (Figure 4A), as well as glomerular damage (Figure 4B), compared to mice from all other groups. This is commensurate with the renal damage found upon histological analysis of renal tissue from MAKI patients [11]. Mice in the 2KI group also showed significantly greater tubular and glomerular damage than those in the non-infected 2KNI group. It is important to point out that more than 50% and 65% of the mice included in this group exhibited minimal to no (score 1 or 0) tubular damage and/or glomerular damage, respectively. In contrast, 10 out of 11 mice in the 1KI group showed at least a score of 2 for tubular damage, and 9 out of 11 for glomerular damage.

We evaluated renal tissue for expression of cleaved caspase-3, a marker of apoptosis, as its expression in the renal tubular cells of MAKI patients has been previously reported [32]. Expression was found to be significantly higher in both 2KI and 1KI mice, compared to control non-infected mice, indicating that apoptosis is occurring in kidneys of all infected mice, regardless of how many kidneys the animals possess.

Our results indicate that this novel experimental mouse model exhibits signs of AKI consistent with those displayed by patients of MAKI and can be used as a standard physiological model to study this SM complication in vivo. All experimental mouse models of SM have limitations and do not reproduce the interactions between infected erythrocytes and endothelial cells, since erythrocytes infected with rodent malaria parasites are not cytoadherent, as the ones infected with the human parasite *P. falciparum* are. In the case of MAKI, there are limited studies on this matter; however, the levels of parasite sequestration in the kidney are relatively low [12]. Very little is currently understood about this pathology, as its prevalence was underestimated until recently. This new model could provide the much-needed foundation upon which to build our understanding of the clinical pathology of MAKI, as well as the underlying molecular mechanisms of this pathogenesis, which can ultimately contribute to the development of urgently needed adjunctive therapies, and identification of early biomarkers for diagnosis.

## 4. Materials and Methods

### 4.1. Animals

Male and female C57BL/6 J mice (10–12 weeks old) were obtained from The Jackson Laboratory (Bar Harbor, ME, USA). All animals were housed in a humidity-controlled (40–60%) and temperature-controlled (21 ± 2 °C) environment with a 12-hr light/dark cycle, and had free access to water and standard diet. All procedures involving animals followed the *Guide for the Care and Use of Laboratory Animals* (NIH Publication No. 85–23, revised 1985) and were performed with the approval of the Animal Care and Use Committee at Lehman College.

Control/experimental groups included: (1) non-infected, non-nephrectomized mice (2KNI); (2) non-infected, nephrectomized mice (1KNI); (3) infected, non-nephrectomized mice (2KI); and (4) infected-nephrectomized mice (1KI). Unilateral nephrectomies were performed on animals in the 1KNI control and 1KI experimental groups. Animals were closely monitored post-operation and allowed ample time (5–6 weeks) to recover from surgery prior to infection. Animals in the 2KI and 1KI experimental groups were infected by ip inoculation with 10^7^ *Pb*NK65-NY infected erythrocytes.

Blood samples (~10 μL) were taken starting at day 3 post infection, and then every other day, via tail bleeds, to create thin blood smears that were then stained with Giemsa (Sigma, St. Louis, MO, USA) to monitor parasitemia. Urine was collected from mice in all groups every 3 days for the duration of 24 h using metabolic cages (Hatteras Instruments, Grantsboro, NC, USA). Animals were euthanized between days 14–15 post-infection.

### 4.2. Histopathological Analysis

Paraffin-embedded 4 μm kidney sections were stained with Gil’s hematoxylin (Sigma, St. Louis, MO, USA) & eosin (Epredia, Kalamazoo, MI, USA), and scanned using the ACCU-SlideMS Manual Slide Scanning System (ACCU-SCOPE, Commack, NY, USA). Scanned images were analyzed using the open software for bioimage analysis, QuPath v 0.2.3. Glomerular and tubulointerstitial damage were assessed using a semi-quantitative scoring method. The total score for glomerular activity was determined by examining renal tissue for glomerular proliferation, karyorrhexis/fibrinoid necrosis, cellular crescents, hyaline deposits, and polymorphonuclear leukocytes. The total score for tubulointerstitial activity was determined by examining renal tissue for tubular cell necrosis/lesions, tubular cell activation, tubular regeneration, and interstitial inflammation.

### 4.3. Quantification of Kidney Injury Biomarkers

Urinary NGAL was quantified in 50 μL of a 1:5000 dilution of urine using the Mouse Lipocalin-2/NGAL DuoSet ELISA kit from R&D Systems (Minneapolis, MN, USA) following the manufacturer’s protocol. Serum Cystatin-C was quantified in 50 μL of 1:1000 dilutions of serum, using the Mouse Cystatin C DuoSet ELISA kit from R&D Systems (Minneapolis, MN, USA) following the manufacturer’s protocol. All samples were measured in duplicate. BUN was quantified in 50 μL of a 1:10 dilution of serum using the Urea Nitrogen (BUN) Colorimetric Detection Kit from Invitrogen (Frederick, MD, USA) following the manufacturer’s protocol. Serum creatinine was quantified in 15 μL of serum using the Creatinine (serum) Colorimetric Assay Kit from Cayman Chemicals (Ann Arbor, MI, USA) following the manufacturer’s protocol. Urinary albumin was quantified in 20–40 μL of urine; Exton solution (Cole-Parmer, Vernon Hills, IL, USA) was added to urine at a 1:1 ratio and samples were incubated for 10 min in the dark. Absorbance was read at 620 nm and values were multiplied by total volume of urine (mL) to determine albumin in mg/day.

### 4.4. Immunoblotting

Animal renal tissues were homogenized in RIPA buffer (Boston BioProducts, Milford, MA, USA) with 1% protease inhibitor cocktail (Sigma, St. Louis, MO, USA) and protein concentrations were quantified using Pierce ™ BCA Protein Assay Kit (ThermoFisher Scientific, Waltham, MA, USA). Laemmli’s sample buffer (Bio-Rad, Hercules, CA, USA) was added to samples and the samples were then boiled at 95 °C for 5 min. Equivalent amounts of protein lysate (30–50 μg) or volumes of urine (15–30 μL) were loaded into SDS/PAGE gels (Bio-Rad, Hercules, CA, USA) and separated proteins were transferred to PVDF membranes (Bio-Rad, Hercules, CA, USA). Membranes were rinsed twice with distilled water, washed three times for 5 min each in TBS-T, then blocked with 5% BSA in TBS-T for 1 hr at room temperature. Membranes were incubated with primary antibodies cleaved caspase-3 diluted 1:1000 (Cell Signaling Technology, Danvers, MA, USA) and GAPDH diluted 1:1000 (Proteintech, Rosemont, IL, USA) overnight at 4 °C. Membranes were washed three times before being incubated for 1 h at room temperature with secondary antibody HRP-conjugated goat anti-rabbit (Cell Signaling Technology, Danvers, MA, USA) diluted 1:2000. Membranes were then washed three times and developed in Pierce™ ECL Western Blotting Substrate (ThermoFisher Scientific, Waltham, MA, USA). Chemiluminescent signals were detected with the BioRad ChemiDoc Imaging System (Bio-Rad, Hercules, CA, USA) and a densitometric analysis of the bands was performed using ImageJ.

### 4.5. Statistical Analysis

All datasets were tested for normal Gaussian distribution using the Shapiro-Wilk test and for homogeneity of variance using the Brown-Forsythe test. Datasets that met the normality and homoscedasticity criteria were further analyzed using parametric ANOVA tests and Dunnett’s T3 tests. Datasets that did not meet the above criteria were analyzed using the Kruskal-Wallis test and Dunn’s tests. All statistical analyses were conducted in R-studio and GraphPad Prism 9.

## Figures and Tables

**Figure 1 pathogens-12-00545-f001:**
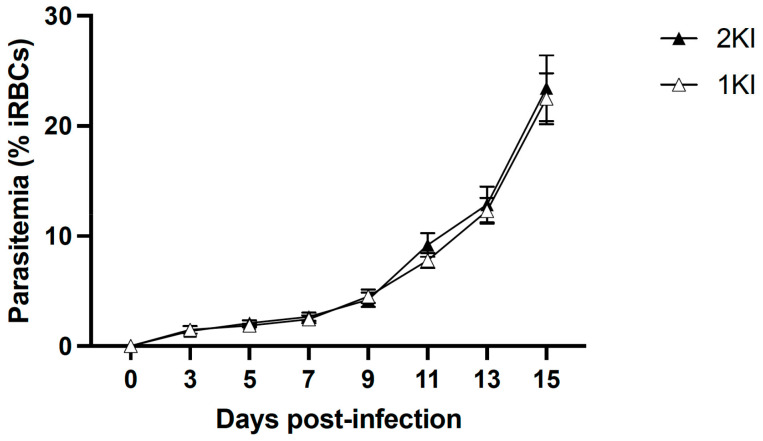
Parasitemia levels. Parasitemia levels in mice were measured every other day, starting from day 3 post-infection, by counting percent of infected red blood cells (%iRBC) using Giemsa-stained thin blood smears. A two-way ANOVA analysis and Sidak’s multiple comparisons tests were performed (ANOVA: *p* = 0.9939, F = 0.1482). Mean parasitemia levels of 1KI (open triangles, n = 10) and 2KI (closed triangles, *n* = 10) mice were comparable from days 0 to 15 post-infection. Results shown as mean ± SEM.

**Figure 2 pathogens-12-00545-f002:**
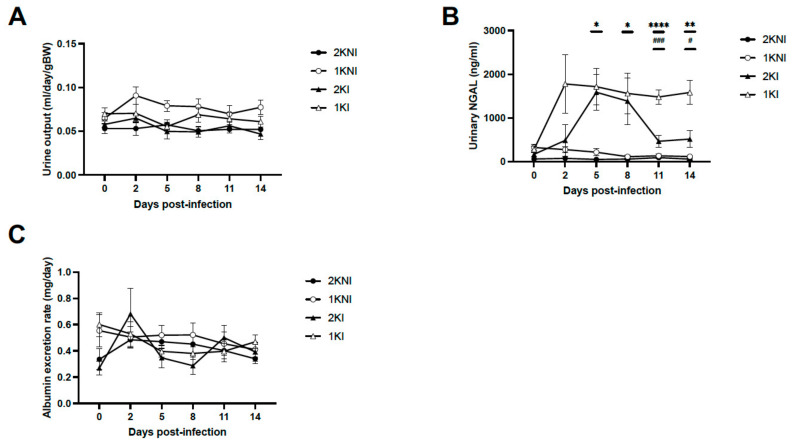
Urinary biomarkers of kidney injury. Urine was collected over the course of 24 h prior to infection and then every 2–3 days post-infection using metabolic cages. (**A**) There were no significant differences in volume of urine output between any group at any day post-infection (ANOVA: *p* = 0.2378, F = 1.250). (**B**) Urinary NGAL (uNGAL) concentrations were measured in ng/mL using an ELISA. uNGAL concentrations of 1KI mice (open triangles, *n* = 10) were significantly higher than those of 2KNI mice (closed circles, *n* = 15) from day 5 post-infection, and significantly higher than those of 2KI mice (closed triangles, *n* = 10) from day 11 post-infection until termination (* *p* < 0.05, ** *p* < 0.01, **** *p* < 0.0001 compared to 2KNI; # *p* < 0.05, ### *p* < 0.001 compared to 2KI). (**C**) Urinary albumin levels were measured using a colorimetric assay. Albumin levels did not differ significantly between groups or increase significantly on any day post-infection (ANOVA: *p* = 0.6639, F = 0.5299). Results shown as mean ± SEM.

**Figure 3 pathogens-12-00545-f003:**
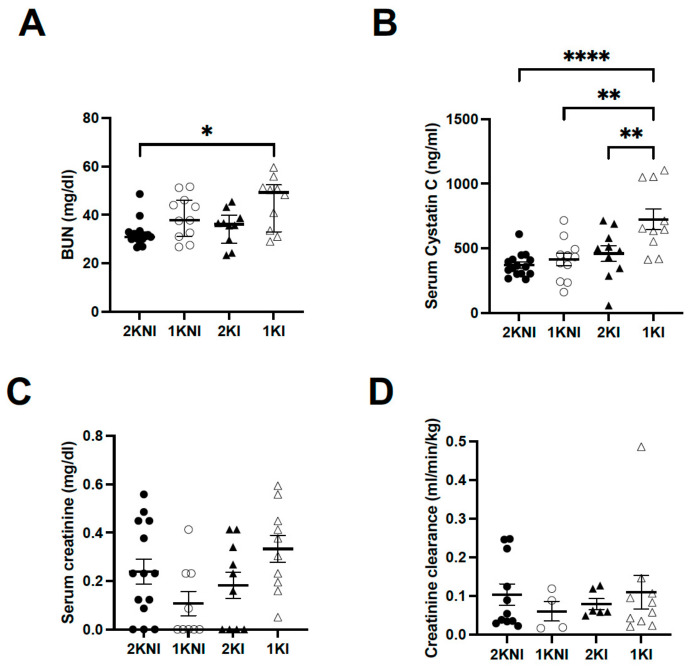
Serum biomarkers of kidney injury. Serum was collected upon termination of model and levels of AKI-specific biomarkers were measured either using colorimetric assays or ELISA. (**A**) Blood urea nitrogen (BUN) levels were significantly higher in 1KI mice (open triangles, *n* = 10) compared to 2KNI mice (closed circles, *n* = 15; * *p* = 0.0118) (Kruskal-Wallis: *p* = 0.0143; H = 10.57). Center lines show the medians; box limits indicate the 25th and 75th percentiles. (**B**) Cystatin C levels were significantly higher in 1KI mice (open triangles, *n* = 10) compared to all other groups (2KNI: closed circles, *n* = 15, **** *p* < 0.0001; 1KNI: open circles, *n* = 11, ** *p* = 0.0012; 2KI: closed triangles, *n* = 10, ** *p* = 0.0090) (ANOVA: *p* = 0.0001; F = 8.853) Results shown as mean ± SEM. (**C**) The mean creatinine level for the 1KI group (*n* = 10, mean = 0.3333) was 1.4-fold higher than that of the 2KNI group (*n* = 14, mean = 0.2391), 3-fold higher than that of the 1KNI group (*n* = 9, mean = 0.1071), and 1.8-fold higher than that of the 2KI group (*n* = 10, mean = 0.1826) but this increase was not of statistical significance (Kruskal-Wallis: *p* = 0.0588; H = 7.452). Center lines show the medians; box limits indicate the 25th and 75th percentiles. (**D**) Creatinine clearance was measured on day 14 and there were no significant differences between any group (Kruskal-Wallis: *p* = 0.6790, H = 1.514). Center lines show the medians; box limits indicate the 25th and 75th percentiles.

**Figure 4 pathogens-12-00545-f004:**
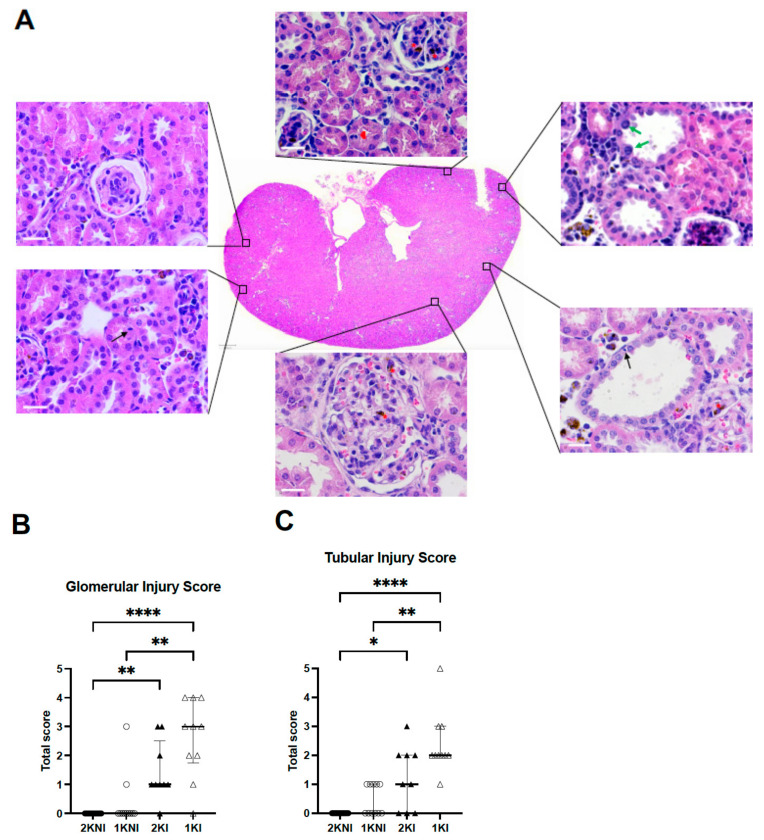
Histological analysis of renal tissue. Histological sections of kidneys from mice at day 14 post-infection were evaluated by a nephropathologist in a blind session for tubular and glomerular damage using a semi-quantitative scoring method with an ascending scale from 0 to 12. (**A**) Representative images of the most common lesions found in nephrectomized infected mice (1KI) are shown at higher magnification. Dilated tubules (top right panel), glomerulomegaly (lower center panel), glomerular endocapillary proliferation (top left panel), tubular apoptotic cells (black arrows), tubular regeneration (red arrowheads), hemosiderin deposits (red asterisks), karyomegaly (green arrows). White scale bars, 20 μm. Black scale bar in central image, 500 μm. (**B**) The mean total score for glomerular injury for 1KI mice (open triangles, *n* = 10) was significantly higher than that of 2KNI (closed circles, *n* = 15, **** *p* < 0.0001) and 1KNI (open circles, *n* = 11, ** *p* = 0.0013) mice. The mean total score for 2KI mice (closed triangles, *n* = 9) was significantly higher than that of 2KNI mice (** *p* = 0.0028), as well. (Kruskal-Wallis: *p* < 0.0001; H = 29.33). (**C**) The mean total score for tubular injury for 1KI mice (open triangles, *n* = 10) was significantly higher than that of 2KNI (closed circles, *n* = 15, **** *p* < 0.0001) and 1KNI (open circles, *n* = 11, ** *p* = 0.0027) mice. The mean total score for 2KI mice (closed triangles, *n* = 9) was significantly higher than that of 2KNI mice (* *p* = 0.0169). (Kruskal-Wallis: *p* < 0.0001; H = 29.55). Center lines show the medians; box limits indicate the 25th and 75th percentiles.

**Figure 5 pathogens-12-00545-f005:**
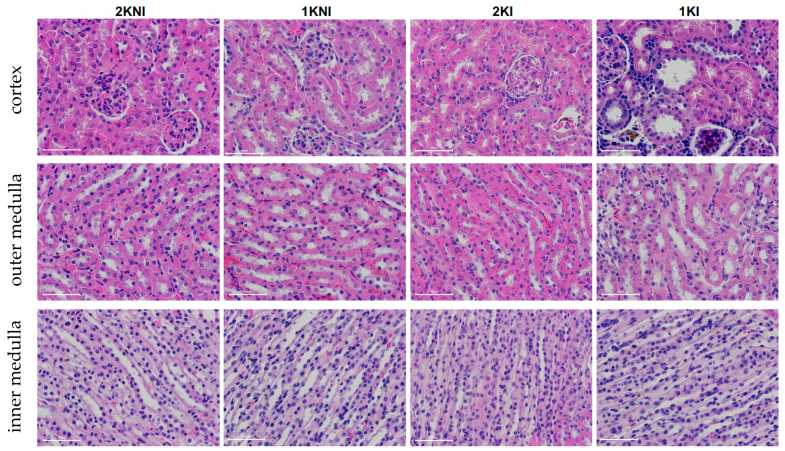
Histological analysis of renal tissue. Histological sections of kidneys from mice at day 14 post-infection showing the cortex (**top** row), the outer medulla (**middle** row) and the inner medulla (**bottom** row). Glomerular and tubular lesions can be observed in the cortex of 1KI group kidney sections and to a lesser degree in the 2KI group. There were minimal to no remarkable changes in the outer medulla or inner medulla in any of the experimental groups included in the study. White scale bars, 50 μm.

**Figure 6 pathogens-12-00545-f006:**
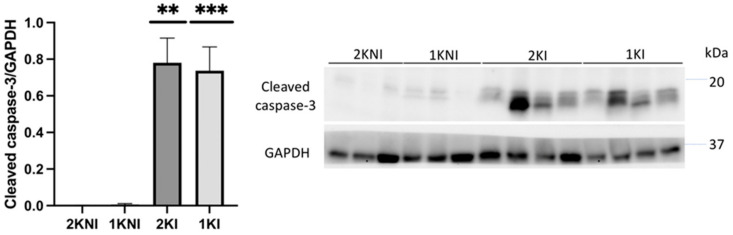
Cleaved caspase-3 expression. Expression of cleaved caspase-3 was detected via immunoblotting and normalized using GAPDH expression. Expression was relatively higher in 1KI (*n* = 9) and 2KI (*n* = 8) mice compared to 2KNI (*n* = 9) and 1KNI (*n* = 9) mice (** *p* < 0.01; *** *p* < 0.001) (Kruskal-Wallis: *p* < 0.0001; H = 27.74).

## Data Availability

Dataset will be available from the study principal investigator on reasonable request (julio.gallegodelgado@lehman.cuny.edu).

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
