# Peer review of "Novel Experimental Mouse Model to Study Malaria-Associated Acute Kidney Injury"

_pathogens, 2023, doi:10.3390/pathogens12040545_

Round 1

Reviewer 1 Report

In this paper authors suggest novel experimental mouse model to study acute kidney injury in malaria. Well written paper with helpful and important Discussion part.

 Minor critics:

·      Fig 1. Data from two groups are reported by triangles, inverted triangles. For distinguish better the data from each group I suggest to use “open” triangles or different colours for groups.

·      Fig 2. Four different conditions are presented with very similar marks on the graph. It’s difficult to distinguish them. Please, use “open”/ ”close” symbols or different colours.

·      Results, lines 117-125. When numerical data for mean values are reported in the text, please add also standard error data in the text.

·      The “score” introduced for the tubular and glomerular injury is important. For 1KI group it is 2.4, but will be useful to know the maximum (or theoretical maximum) of such “score” which permits to understand the “gravity” of observed scores for 1KI. Which score authors suppose for worst kidney injury (injuries not compatible with life)?

Author Response

Thank you for reviewing our manuscript. We appreciate the thoughtful feedback and constructive criticism. We have included the following changes as suggested.

  • Figures: The symbols used to represent the four different conditions in the study have been changed according to the reviewer’s suggestion to open/close symbols. They are now more easily distinguishable. These have been adjusted in all figures and legends.
  • Results: Lines 119-129 have been adjusted to include mean plus standard error for each group.
  • Scores for tubular/glomerular injury: To address this, we introduced the scale used for the score in the text (line 149) as well as in the legend of figure 4.

Reviewer 2 Report

Manuscript pathogens-2306240 entitled "NOVEL EXPERIMENTAL MOUSE MODEL TO STUDY MALARIA-ASSOCIATED ACUTE KIDNEY INJURY" shows to develop an in vivo model of Malaria-induced acute kidney injury using unilateral nephrectomy plus infection with Plasmodium berghei NK65.  However, it is hard to say that malaria induces "ACUTE" kidney injury in mice, even though showing some functional and structural damages.  The authors should consider the following major points:

1. Renal function. 

Please check GFR or creatinine clearance at 14 days after infection.  Actually, the authors should measure creatinine concentration in blood and urine at 2, 5, or 8 days because 14 days are enough time to restore renal dysfunction.  I know the authors already measured urinary NGAL for 14 days, but as you know, restored kidneys can still reveal high urinary NGAL/sodium/etc excretion.  Furthermore, GFR and serum/plasma creatinine are crucial markers of AKI in clinical and animal-experimental fields.

Taken together, the authors should measure 1) GFR or creatinine clearance at 14 days and if not significant, 2) either parameter during early period (2, 5 or 8 days).

2. Structural damage.

Please show tubular injury index and other representative picture.  Probably, authors-indicated dilated tubule in right panels of figure 4 is not dilated.  This is virtually a normal collecting tubule or connecting tubule.  Consistent with functional data, the authors should measure tubular injury score at 2, 5 or 8 days after infection.  If it's not possible to show a signficantly increased tubular injury score, it's difficult to demonstrate malaria-induced AKI.  Furthermore, which region of the kidney is more susceptible to malaria-induced kidney injury.  At least, the representative pictures in the cortex and the outer/inner medulla in all groups should be revealed.  For details please see and cite: https://www.mdpi.com/2072-6651/15/2/118 and https://link.springer.com/article/10.1007/s43440-022-00403-x

Taken together, the authors should measure 1) tubular injury score at 14 days, if not significant, 2) the score during early period (2, 5 or 8 days), and finally show the representative pictures in the cortex and the outer medulla of mice in all groups.

3. Graph.

Non-parametic data should be presented as median value with quartiles, not mean plus/minus SEM.  Please introduce if the data in respective panels of all figures were analyzed using ANOVA or Kruskal-Wallis tests.  Also, provide the p and F values of ANOVA and H value of Kruskal-Wallis in each figure caption.  Lastly, legends in graphs are more helpful to read figures.

I think the manuscript clearly deserves publication and will interest a broad readership.  My main cocnern is not about the idea, which is very nice, but more about the experimental design and results, which are less to involve to AKI.  However, the current data seems to be connected to "mild kidney injury" in mice.

Author Response

Thank you for reviewing our manuscript. We appreciate the thoughtful feedback and constructive criticism and have tried to address each critique accordingly.

  1. Renal function

We agree with the reviewer that GFR and serum/plasma creatinine are crucial markers of AKI in clinical and animal-experimental fields. As per the reviewer’s suggestion, we measured creatinine clearance at day 14 and there was no significant difference among groups. We agree that measuring serum biomarkers at multiple days throughout the study could have provided valuable information, however, the amount of blood required for these measurements (~100ul/animal/day) would have exceeded the limits allowed by our animal welfare committee. The alternative would have been to increase the number of animals in the study to at least 50/group totaling 200 mice, which is why we measured creatinine only at the endpoint. We do acknowledge that creatinine clearance is widely used as a marker for AKI in the field, however, in the context of malarial infection, these biomarkers may not be as accurate in detecting early stages of AKI, as are serum Cystatin C and NGAL (see main text lines 252-256).

  1. Structural damage

We apologize for the poor choice of words used in figure 4 and the corresponding text when referring to the tubular and glomerular injury. As suggested by the reviewer, we have fixed this by rewording the text to include “tubular injury” and “glomerular injury” to clearly reflect the fact that tubular and glomerular injury were assessed at day 14 and that the tubular injury score was significantly higher in the nephrectomized-infected (1KI) group. As per the reviewer’s suggestion, we have replaced one image with another showing clearer instances of tubular dilation (top right panel). The image in the lower right panel has been kept in order to show the instance of apoptotic nuclei but the mention of tubular dilation was removed as it was not so clear. As requested, we have included a supplemental figure inspired by the figures found in the articles referenced by the reviewer with representative images from the cortex, inner, and outer medulla in mice from all groups side by side for comparison (supplemental figure 2). This has also been mentioned in the main text (lines 159-161).

  1. Graphs

Graphs have been adjusted so that non-parametric data is presented with median value with quartiles. The figure legends have also been adjusted to include p and F/H values for ANOVA and Kruskal-Wallis tests, respectively. Symbol legends have also been added to the figures to make them clearer.

Round 2

Reviewer 2 Report

Please add the supplementary data to the main text.  Don't hide them. 

The others are okay. 

Author Response

Dear reviewer,

Supplemental figures have been integrated into the main text as requested, and now they are Figure 4D for the creatinine clearance and Figure 5 for the histology of the cortex, outer and inner medulla.